# Enhancing Bioactive Compound Extraction from Rose Hips Using Pulsed Electric Field (PEF) Treatment: Impacts on Polyphenols, Carotenoids, Volatiles, and Fermentation Potential

**DOI:** 10.3390/molecules30153259

**Published:** 2025-08-04

**Authors:** George Ntourtoglou, Chaido Bardouki, Andreas Douros, Nikolaos Gkanatsios, Eleni Bozinou, Vassilis Athanasiadis, Stavros I. Lalas, Vassilis G. Dourtoglou

**Affiliations:** 1Department of Wine, Vine and Beverage Sciences, University of West Attica, Agiou Spyridonos, 12243 Egaleo, Greece; gntourtoglou@uniwa.gr; 2Vioryl S.A., 28th km National Road Athens, Lamia, 19014 Afidnes, Greece; bardouki@vioryl.gr (C.B.); adouros@vioryl.gr (A.D.); wvbs20685014@uniwa.gr (N.G.); 3Department of Food Science and Nutrition, University of Thessaly, Terma N. Temponera Street, 43100 Karditsa, Greece; empozinou@uth.gr (E.B.); vaathanasiadis@uth.gr (V.A.)

**Keywords:** rose hips, tiliroside, robinin, *Torulaspora delbrueckii*, GC-MS analysis, fermentation dynamics, polyphenol extraction, PEF extraction

## Abstract

Rose hips are rich in polyphenols, making them a promising ingredient for the development of functional fruit-based beverages. This study aimed to evaluate the effect of Pulsed Electric Field (PEF) extraction treatment on rose hip (RH) pulp to enhance the extraction of polyphenols, carotenoids, and volatile compounds. Additionally, this study examined the impact of adding rose hip berries during different stages of carbohydrate fermentation on the resulting phenolic and aroma profiles. A control wort and four experimental formulations were prepared. Rose hip pulp—treated or untreated with PEF—was added either during fermentation or beforehand, and the volatiles produced were analyzed using GC-MS (in triplicate). Fermentation was carried out over 10 days at 20 °C using *Saccharomyces cerevisiae* and *Torulaspora delbrueckii*. At a 10:1 ratio, all beverage samples were subjected to physicochemical testing and HPLC analysis for polyphenols, organic acids, and carotenoids, as well as GC-MS analysis for aroma compounds. The results demonstrated that the use of PEF-treated rose hips significantly improved phenolic compound extraction. Moreover, the PEF treatment enhanced the aroma profile of the beverage, contributing to a more complex and appealing sensory experience. This research highlights the rich polyphenol content of rose hips and the potential of PEF-treated fruit as a natural ingredient to improve both the functional and sensory qualities of fruit-based beverages. Their application opens new possibilities for the development of innovative, health-promoting drinks in the brewing industry.

## 1. Introduction

Roses are known not only for their romantic significance but also for their contributions to medicine, cosmetics, and dietary supplements. Members of the Rosaceae family are well known for their antioxidant properties and their abundance of vitamins, particularly vitamin C [1]. Roses, with their diverse species, offer a wealth of culinary and medicinal possibilities. The edible parts include rose hips, the fruit that develops after the rose flower has bloomed. Many species of *Rosa*, including the Dog Rose (*R. canina*), Ramanas Rose (*R. rugosa*), and French Rose (*R. gallica*) produce edible hips that can be used in a variety of ways. From teas to jams, desserts to garnishes, jellies, syrups, stews, sauces, and soups, these roses are valued for their versatility and health benefits.

Rose hip (RH) extract has been investigated for its potential effects on low-density lipoprotein cholesterol (LDL-C) and cardiometabolic health. A 2023 systematic review by Belkhelladi [2] concluded that RH supplementation may reduce LDL-C and total cholesterol, particularly in individuals not using statins, while also highlighting its favorable safety profile. A randomized, double-blind, crossover trial by Andersson et al. [3] demonstrated that daily intake of 40 g of RH powder over six weeks significantly reduced LDL-C (−6%) and systolic blood pressure (−3.4%) in obese individuals. A small-scale study by Marstrand et al. [4] observed metabolic improvements in osteoarthritis patients supplemented with RH powder, with more pronounced effects in lean individuals. Additionally, Dabaghian et al. [5] found that RH extract significantly reduced fasting blood glucose in type 2 diabetic patients after three months of treatment.

Secondary benefits include reductions in blood pressure and body weight, as seen in both the Andersson et al. [3] trial and a study by Nagatomo et al. [6], which showed that 100 mg/day of RH extract significantly reduced visceral fat and body mass index (BMI) in pre-obese Japanese adults. Mechanistically, RH polyphenols have been reported to modulate peroxisome proliferator-activated receptor (PPARα) signaling and enhance fatty acid oxidation—findings echoed in animal studies such as Cavalera et al. [7], which demonstrated the activation of thermogenic pathways in ApoE knockout mice.

Rodent data suggest dose-dependent effects, with 250 mg/kg providing optimal benefits in models of metabolic syndrome [6]. Nonetheless, the variability in formulations, dosages, and populations studied contributes to mixed results in human trials. Overall, RH remains a promising complementary therapy for cardiometabolic risk reduction, particularly in individuals at an increased risk of type 2 diabetes or cardiovascular disease.

Roses are rich in various polyphenols, including flavanols, flavonols, and flavanes, which are found in different *Rosa* spp. genotypes [8,9]. Tiliroside (*cis* and *trans*) and robinin, a glycosyloxyflavone, are among the most biologically significant compounds in roses.

Extracts produced by traditional extraction methods, such as those using 80% aqueous acetone from *R. canina* seeds, have demonstrated the ability to inhibit weight gain and visceral fat in mice without affecting food intake [10]. *Trans*-tiliroside, the primary component of the extract, appears to be the active ingredient, significantly reducing weight gain (particularly visceral fat) and lowering plasma glucose after glucose loading [10]. The kaempferol and *p*-coumaroyl portions of tiliroside are both crucial for these anti-obesity effects. Tiliroside may even outperform orlistat, highlighting its potential for the development of new anti-obesity drugs. Orlistat is a lipase inhibitor commonly used to reduce dietary fat absorption and serves here as a comparative benchmark for lipase inhibitory activity.

Concerning extraction methods, emerging green techniques like Pulsed Electric Field (PEF) offer superior extraction capabilities. PEF is a non-thermal method that utilizes pulsed electrical fields to permeabilize cell membranes, enhancing the extraction of aromatic and flavor compounds [11]. As indicated by our previous studies, PEF treatment can improve the mass transfer of intracellular components, facilitating the release of valuable aromatic and phenolic compounds and pigments in the food and pharmaceutical industries [12,13].

Numerous previous studies have demonstrated positive results of PEF in the treatment of apples, grapes, hop pellets, and other plant materials [14]. In fact, studies on members of the Rosaceae family have shown that PEF application significantly reduces distillation time [11]. Additionally, PEF has been applied to hops, enhancing the extraction of essential bitter compounds and bioactive ingredients used in brewing, thereby improving process efficiency and product quality [13]. Furthermore, recent research has explored the application of PEF and treatment with small scrap wood pieces in the aromatization of red wine, demonstrating its potential to improve the extraction of bioactive compounds [15,16].

The fermentation of RH has been widely explored due to the fruit’s high content of bioactive compounds, natural sugars, and organic acids, which contribute to the production of aromatic and flavorful fermented products. Recently, Tsapou et al. [17] proved that the application of PEF to RH pulp before fermentation enhances the release of intracellular metabolites, improving microbial activity and fermentation efficiency. This confirms PEF as a promising pretreatment technique for optimizing the biochemical profile of fermented RH products.

Furthermore, fermentation processes can be strategically employed to not only produce alcoholic beverages but also to enhance the extraction of valuable compounds from RH [18]. The alcohol generated during fermentation acts as an efficient solvent [19,20,21], increasing the solubility of flavonols and polyphenols, which are known to possess various health benefits. The synergistic effect of fermentation and alcohol-mediated extraction can lead to products with an enhanced bioactive content and an improved bioavailability of key compounds. *Torulaspora delbrueckii*, a yeast known for its ability to ferment with a low alcohol production, can be employed to retain delicate flavors and aromas while still facilitating the extraction of beneficial compounds from rose hips [22].

The aim of this study was to investigate the effects of PEF treatment on RH with regard to lycopene, β-carotene, and tiliroside—a glycosyloxyflavone derived from kaempferol. Finally, the effect on the fruit’s physical properties (e.g., water content) was determined. The sugar content was also measured to assess the impact of PEF on fermentation efficiency. To the best of our knowledge, no previous studies have explored the possibility of extracting these compounds using PEF.

## 2. Results and Discussion

### 2.1. Moisture Content

The moisture content was calculated as 21.3%, based on the AOAC Official Method 925.10 [23]. The reported water content refers to all starting samples to ensure consistent comparisons across experimental conditions.

### 2.2. Extraction of Carotenoids

The results of carotenoid extraction shown suggest that PEF treatment affects carotenoid extraction differently depending on the compound.

The PEF-treated samples exhibited a significantly higher lycopene concentration (0.029 mg of lycopene per g dry weight (DW)) compared with the control (0.014 mg of carotenoid per g DW), indicating that the technique enhances this specific compound’s release from RH. This was statistically significant (*p* < 0.05). This may be attributed to PEF-induced cell membrane permeabilization, which facilitates the diffusion of intracellular compounds. Lycopene—a highly hydrophobic carotenoid—is typically bound within lipid-rich cellular structures, and PEF likely disrupts these compartments, improving extraction efficiency.

By contrast, the β-carotene levels remained similar between the control and PEF-treated samples (0.078–0.079 mg of β-carotene per g DW), suggesting no statistically significant difference. This may indicate that β-carotene is less affected by PEF or that the extraction method already achieves near-maximal recovery under both conditions. Its stability and localization in plant matrices could also contribute, as β-carotene may reside in structures less susceptible to PEF disruption than those harboring lycopene.

In summary, PEF treatment enhances lycopene extraction but has little effect on β-carotene release. This selective improvement in carotenoid recovery highlights PEF’s potential for optimizing bioactive compound extraction, particularly in lycopene-rich functional foods and beverages. The efficiency of the PEF-assisted extraction of carotenoids, particularly lycopene, has been supported by previous studies. Pataro et al. [24] demonstrated that applying PEF (5 kV/cm, 5 kJ/kg) to tomato peels significantly enhanced lycopene extraction yields by 12–18% and increased the extraction rate by up to 37% compared with untreated samples, without causing the isomerization or degradation of the compound. Similarly, Luengo et al. [25] reported a substantial improvement in carotenoid recovery from tomato peels using PEF (5 kV/cm, 90 µs), with lycopene yields increasing from 38 µg/g FW in untreated samples to 58 µg/g FW post treatment. These findings align with our results, confirming that PEF promotes the electroporation of plant cell membranes, thereby facilitating the release of intracellular compounds such as lycopene while preserving their chemical integrity. Further studies could explore how PEF intensity, treatment duration, and solvent composition affect the efficiency of carotenoid extraction.

### 2.3. Volatile Compounds After Fermentation

#### 2.3.1. Volatile Profile of Fermented Samples

The analysis of volatile compounds in fermented RH samples (Table 1) indicates that co-culturing *Saccharomyces cerevisiae* and *Torulaspora delbrueckii* in a 1:10 ratio resulted in a rich aromatic profile. This part of the study assessed whether PEF treatment could improve aroma development via the enhanced extraction of amino acids and other yeast metabolites.

The PEF-treated samples exhibited slightly higher concentrations of isoamyl alcohol—approximately 1 mg/L more than the control—although the difference was not statistically significant. Phenylethyl alcohol, however, a compound responsible for a rose-like aroma, was significantly increased in the PEF-treated samples (8.85 ± 0.76 mg/L vs. 5.96 ± 0.90 mg/L).

In contrast, nonanoic acid, α-ionone, and eugenol were unaffected, indicating that PEF treatment does not influence all volatiles equally. The most notable effects were observed in medium-chain fatty acids (MCFAs), tryptophol, and phenylethyl alcohol—key metabolites of the Ehrlich pathway.

Beyond enhancing the mouthfeel and sensory experience, MCFAs are precursors for MCFA esters, which contribute fruity and less solvent-like aromas. These findings suggest that PEF may be a valuable tool for optimizing the floral and fruity sensory profile of rose hip-based fermented beverages.

#### 2.3.2. Compound-Specific Effects of PEF on Volatiles

The comparison of chemical concentrations between the control and PEF treatments after fermentation (Table 2, Table 3 and Table 4) revealed distinct and statistically meaningful patterns. Notably, phenylethyl alcohol increased by 48.5%, tryptophol by 130%, and n-hexadecanoic acid by 69.8% in the PEF-treated samples. Conversely, vanillic acid and 3-decenoic acid showed marked decreases of 61.8% and 57.1%, respectively. Intriguingly, a novel compound—1H-indole-3-ethanol acetate—was detected exclusively in the PEF-treated samples, suggesting the activation of previously dormant metabolic pathways.

The impact of PEF treatment varied across chemical categories: aromatic compounds and long-chain fatty acids generally increased, whereas simple alcohols and short-chain acids remained relatively unchanged. For example, butanoic acid levels stayed constant (0.11 mg/L) in both conditions, and isoamyl alcohol exhibited minimal variation. These findings imply that PEF treatment selectively influences complex molecular biosynthesis without disrupting simpler structural components.

Statistical reliability was high for key metabolites such as phenylethyl alcohol, tryptophol, and vanillic acid, as indicated by relatively low standard deviations. Some volatiles exhibited greater variability, reflecting the nuanced influence of PEF on fermentation-derived metabolite pathways. Medium-chain fatty acids (MCFAs), in particular, emerged as a focal point, serving not only as precursors for fruity MCFA esters but also as drivers of improved mouthfeel and sensory perception.

This evidence strongly supports the use of PEF as a targeted tool for enhancing aromatic complexity and selectively modulating fermentation chemistry. When combined with co-cultures of *T. delbrueckii* and *S. cerevisiae* (at a 1:10 ratio), the fermentation of RH produced beverages with elevated floral and fruity notes, balanced acidity, and a notable retention of bioactive compounds—while simultaneously reducing ethanol content. This synergistic integration of non-*Saccharomyces* yeast fermentation and PEF pretreatment offers a compelling approach to developing functional, probiotic-rich, low-alcohol beverages with superior nutritional and sensory qualities.

### 2.4. Sugar Content Analysis

As illustrated in Figure 1, the PEF-treated rose hip samples showed a clear increase in both fructose and glucose levels compared with the control. UHPLC-MS analysis revealed that the peak areas for both sugars were notably elevated, indicating that PEF technology may enhance sugar extractability from RH tissues. Most strikingly, the glucose content in the PEF-treated sample more than doubled. Specifically, for glucose there was a significant increase (*p* < 0.05) from 16 g/kg (CTRL) to 34 g/kg (PEF), while for fructose it was from 22 g/kg (CTRL) to 28 g/kg (PEF). These results suggest a significant influence of PEF on intracellular sugar release.

This enhancement is likely a result of PEF-induced cell membrane permeabilization, which facilitates the diffusion of intracellular metabolites. While the exact mechanism remains to be fully elucidated, the data support the hypothesis that PEF creates micro-disruptions in the cell walls, promoting sugar migration into the solvent phase. Further studies are warranted to optimize PEF parameters—such as field strength, pulse duration, and solvent composition—to maximize sugar recovery for functional beverage development.

### 2.5. The Effect of PEF on Catechin, Epicatechin, Tiliroside, and Robinin

Figure 2 and Figure 3 display UV chromatograms, base peak chromatograms (BPC), and mass spectra highlighting catechin, epicatechin, tiliroside, and robinin. These compounds have been well documented in *Rosa* spp. and acknowledged for their bioactivity and health-promoting properties. The chromatograms were generated from samples treated only with water, both with and without PEF exposure.

In comparison, the PEF-treated samples exhibited a significantly higher number of detectable phenolic compounds than their untreated counterparts. This effect was particularly pronounced during the “break periods” in RH samples immersed in clear water, where the disintegration of cellular structures by PEF allowed deeper penetration and extraction.

Further enhancement was observed in samples pretreated with a neutral aqueous solution containing trace amounts of 0.1 N KOH. This alkaline pretreatment facilitated the hydrolysis of ester bonds and the partial breakdown of the cellular matrix, enabling the more efficient release of bound phenolic compounds. Importantly, this increase in yield was consistent across subsequent processing steps—including fermentation and PEF treatment—indicating that initial chemical pretreatment plays a dominant role in extraction efficiency.

The synergy between alkaline hydrolysis and physical cell disruption under PEF appears to substitute for more aggressive acidic or basic extraction protocols, offering a gentler and more environmentally friendly alternative. Fermented or unfermented, treated or untreated, samples consistently demonstrated elevated recovery rates of phenolic compounds, validating the approach as both effective and demonstrating potential scalability. Pilot-scale trials will be necessary to confirm performance at larger volumes.

Figure 4 presents representative MS chromatograms (BPC) comparing the control (blue line) and PEF-treated (red line) samples. The PEF-treated samples revealed a more complex profile, with an abundance of unique MS fragmentation patterns. Many of these peaks correspond to metabolites such as catechin, epicatechin, and tiliroside; others, as detailed in Table 5, represent unknown constituents. Tentative identifications were made using publicly accessible electronic databases available online [26,27,28] and the existing literature, and according to their molecular weight indication. The constituents are presented according to the peak number from Figure 4 (red line, PEF-treated sample). Some peaks have multiple possible compounds due to their similar masses. Peaks with no suggestions were not identified.

### 2.6. Identification and Significance of Key Phenolic Compounds

Catechin and epicatechin are well-known flavan-3-ols commonly found in plant extracts and widely studied for their potent antioxidant activity. Chromatographic analysis revealed elution peaks between 15 and 20 min (HPLC analysis), consistent with reports in the literature for these compounds using reverse-phase HPLC [31,32]. Mass spectrometric data confirmed the presence of catechin and/or epicatechin, showing deprotonated molecular ions [M−H]^−^ at a mass-to-charge (*m*/*z*) of 289.5. Furthermore, *m*/*z* 579 is also observed corresponding to the [2M−1].

Multiple signals sharing the same mass suggest that both structural isomers—catechin and epicatechin—may coexist. While definitive identification would require further MS/MS fragmentation or NMR analysis, their presence aligns with prior findings in *Rosa* spp., where these flavonoids contribute significantly to antioxidant potential and therapeutic effects [33].

Tiliroside, a glycosylated flavonoid, was detected at a retention time of ~35 min (HPLC analysis)—later than catechins, which is typical for flavonoid glycosides due to their increased polarity. The MS spectrum exhibited a clear signal at *m*/*z* 593.5 ([M − H]^−^), confirming tiliroside’s identity, consistent with previous findings in rose hip extracts [34,35]. Tiliroside is associated with a broad spectrum of bioactivities, including antioxidant, anti-inflammatory, antimicrobial, and metabolic effects [36,37]. In this study, its chromatographic peak was relatively isolated, minimizing co-elution with other flavonoids and enhancing the potential for accurate quantification and purification.

Robinin, another glycosyloxyflavone, was also identified in the rose hip samples and has previously been reported in *R. canina* and *Robinia pseudoacacia*. Structurally, robinin is a kaempferol derivative substituted with a complex glycosylation pattern: a 6-O-(6-deoxy-α-L-mannopyranosyl)-β-D-galactopyranosyl residue at position 3, and a 6-deoxy-α-L-mannopyranosyl residue at position 7. Functionally, robinin contributes to the antioxidant and anti-inflammatory properties of rose hips, with reported cardioprotective and neuroprotective effects [38].

Together, these flavonoids—catechin, epicatechin, tiliroside, and robinin—represent the key bioactive constituents of *R. canina*, contributing not only to its antioxidant strength but also to its potential role in metabolic health, anti-aging formulations, and functional beverage design. Their identification further underscores the value of PEF-enhanced extraction and targeted analytical profiling for revealing complex phytochemical landscapes in medicinal plants.

### 2.7. Analytical Observations

The chromatographic separation and mass spectrometric data confirmed the presence of several key bioactive flavonoids—namely catechin, epicatechin, tiliroside, and robinin—in the rose hip extract. Their distinct retention times and *m*/*z* values are consistent with the existing literature, validating the robustness of the analytical approach.

Catechin and epicatechin were detected as monomers (*m*/*z* 289.5) The presence of both structural isomers remains highly probable, though further confirmation via MS/MS or NMR is recommended.Tiliroside exhibited a clear signal at *m*/*z* 593.5 with a well-resolved peak, indicating its substantial abundance and importance within the flavonoid profile.Robinin, also present, reinforces rose hips’ therapeutic potential, given its antioxidant, anti-inflammatory, and cardioprotective roles.

These findings reinforce rose hips as a rich reservoir of pharmacologically relevant flavonoids. Future structural elucidation could provide deeper insights into compound diversity and optimize extraction strategies for nutraceutical applications.

In Table 6, a brief characterization of noteworthy compounds identified in the extract, organized by biological relevance, are presented.

Multiple entries (e.g., Phloridzin, Maclurin derivatives) were observed across several peaks, suggesting either repeated detection or co-elution phenomena. Some compounds—such as 6′’-O-Acetylglycitin and complex quercetin glycosides—highlight structural diversity and point toward modified bioavailability and metabolism.

The diverse presence of flavonoids, tannins, phytoestrogens, and terpenes in RH underscores their powerful nutraceutical potential. From antioxidant defense to cardiometabolic modulation, these constituents may support the development of functional foods and herbal therapeutics tailored to preventive health and wellness.

### 2.8. Heat Map Analysis and Extraction Dynamics

The heat maps presented in Figure 5 and Figure 6 visualize the molecular abundance variations in RH extracts across different treatment conditions: control vs. PEF and fermented vs. non-fermented, using water and water–ethanol solvent systems, respectively. These graphical representations reveal distinct trends in the bioactive compound profiles, helping to interpret how fermentation and Pulsed Electric Field (PEF) treatment jointly shape the chemical landscape.

#### 2.8.1. Key Observations from Water-Based Extracts

The total molecular abundance was significantly reduced following fermentation across both the control and PEF samples. Specifically, the non-fermented control and PEF samples exhibited total abundances of approximately 7.97 × 10^7^ and 7.55 × 10^7^, respectively, while the fermented counterparts dropped to 3.31 × 10^7^ and 2.70 × 10^7^. This demonstrates that fermentation substantially depletes phenolics, flavonoids, and organic acids in aqueous extracts.

Interestingly, PEF treatment did not drastically alter the total abundance compared with the control under either fermentation condition, suggesting a limited impact on overall extraction yield in pure water. However, compound-specific differences were noted. For instance, epicatechin (*m*/*z* 289) showed an increased abundance in the PEF-treated non-fermented samples, suggesting the enhanced extractability of select molecules.

Citric/quinic acid (*m*/*z* 191) and quercetin derivatives (*m*/*z* 431.2 and 447.2) showed significant declines post fermentation. These shifts suggest fermentation’s dominant role in modifying acid and flavonoid levels, potentially due to microbial metabolism or degradation.

#### 2.8.2. Insights from Hydroalcoholic Extracts

Extraction using a 50% water/ethanol solvent improved overall compound recovery. The non-fermented control samples yielded 8.93 × 10^7^ in total abundance, compared with 2.68 × 10^7^ post fermentation. Similarly, the PEF-treated samples decreased from 8.47 × 10^7^ to 3.59 × 10^7^ with fermentation, reaffirming fermentation’s suppressive effect on extractable compounds even in a polar solvent system.

PEF exhibited a more pronounced influence in hydroalcoholic media. The following observations were notable:Phloridzin and taxifolin pentoside (*m*/*z* 435.1) showed a 4.5× increase with PEF treatment in non-fermented samples.Macrulin 3C (*m*/*z* 695.5) abundance rose significantly in the fermented PEF samples, compared with the fermented controls, suggesting the selective enhancement of complex phenolics.

These findings highlight how solvent polarity amplifies PEF’s extraction benefits, particularly in recovering specific bioactives post fermentation. While fermentation remains the primary determinant of compound depletion, PEF treatment provides a targeted advantage for key metabolites.

#### 2.8.3. General Trends from Heat Map Data

Fermentation consistently lowers the abundance of acids and flavonoids, regardless of the extraction method.PEF selectively boosts the extraction of certain phenolic compounds, mainly in non-fermented conditions.The combination of fermentation and PEF yields the lowest total abundances, suggesting fermentation overrides PEF’s extraction advantage.Hydroalcoholic solvents outperform water alone in recovery efficiency, especially when paired with PEF.

These patterns suggest that optimizing solvent composition and pretreatment strategies can significantly influence the recovery of desired bioactive compounds. This nuanced understanding supports tailored extraction protocols for functional beverages, phytochemical profiling, or pharmaceutical applications.

Two additional mechanisms help explain the key observations. First, ethanol produced during fermentation acts as a solvent, improving the extractability of phenolic compounds; however, microbial enzymes simultaneously degrade or transform these molecules, resulting in a net reduction in measurable content. Second, the more pronounced effect of PEF in hydroalcoholic systems is due to the intermediate polarity of ethanol, which enhances solubility. This synergistic effect with PEF-induced membrane permeabilization leads to greater compound recovery than in water alone.

## 3. Materials and Methods

### 3.1. Materials

#### 3.1.1. Rose Hip Fruits

All samples were harvested from Vardousia Mountain, Greece (38°40′52″N 22°08′33″E). The species of rose hip fruit used was *Rosa canina*. The harvested rose hip fruits were transferred to the laboratory and stored at 4 °C until analysis. All analyses were performed within three weeks.

#### 3.1.2. PEF Equipment

The PEF equipment used was previously described by Tsapou et al. [17]. The batch processing chamber (TC) was specifically constructed for the experiment. The device consisted of two stainless-steel plates (5 mm thick, 100 mm in width and length), insulated with Teflon bars (10 mm wide and 10 mm long) along the inner edge. Each stainless-steel plate had a special port for the wires, positioned 5 mm from the edge.

The method for evaluating the electric field strength (*E*) was previously described by Ntourtoglou et al. [13]. For GC-MS (Agilent Technologies Inc., Santa Clara, CA, USA) and UV–Vis (Shimadzu Corporation, Kyoto, Japan) analysis, tap water was used as the extraction solvent. The electric field strength was *E* = 1 kV/cm, with a treatment duration of 30 min (*t*_i_ = 1 ms, *t*_p_ = 1 s, 1800 pulses). There was no dielectric breakdown since the IGBT voltage limit was set at 1100 V. The pulse width was approximately 1 ms, and the pulse frequency was 1 Hz. The total treatment time was 75 ms, and the temperature increase caused by the treatment was negligible (<1 °C).

#### 3.1.3. Yeast Strains

One commercial strain of *Saccharomyces cerevisiae* S.c. US-05 (Fermentis by Lesaffre, Marcq-en-Baroeul, France) and one strain of *Torulaspora delbrueckii* Prelude (Hansen, Hørsholm, Denmark) were used. All strains were hydrated prior to inoculation into the medium.

#### 3.1.4. Chemicals

Diethyl ether (95%), pentane (95%), dichloromethane (95%), *n*-hexane (95%), ethyl acetate (95%), ethanol (HPLC grade, 99%), water (HPLC grade), and anhydrous sodium sulfate were purchased from Chem. Lab (Athens, Greece). The sugars, namely glucose, fructose, and maltose, were of 98% purity (Sigma-Aldrich, Steinheim, Germany). The remaining constituents (yeast extract, (NH_4_)_2_SO_4_, MgSO_4_, ZnSO_4_, KH_2_PO_4_, and KHPO_4_) were obtained from Merck KGaA (Darmstadt, Germany).

### 3.2. Methods

To assess the influence of pH and fermentation on the stability of non-volatile compounds of RH, an extraction plan was implemented (see Figure 7). The protocol involved two primary approaches: (i) the direct dissolution of RH in distilled water under various treatment conditions, (ii) alkaline pretreatment using potassium hydroxide (KOH, 1 N) prior to dissolution.

Each RH pulp sample was 200 mL (100 g pulp in 200 mL solvent) and was subjected to combinations of processing variables, including solvent type (H_2_O or H_2_O:EtOH 50% *v*/*v*), physical processing through PEF or control conditions (CTRL), and optional fermentation (F). These permutations—with and without KOH pretreatment—allowed for the evaluation of individual and synergistic effects of extraction parameters on compound stability and recovery efficiency. PEF was applied to RH pulp dispersed in either water or water–ethanol mixtures.

Table 7 presents the experimental conditions for rose hip treatment groups categorized by solvent, pretreatment, processing (CTRL or PEF), and fermentation (F). The table outlines all tested combinations, enabling the comparative analysis of treatment effects on compound stability and extract yield.

#### 3.2.1. Moisture Content

Moisture was calculated via the AOAC Official Method 925.10 [23]. Specifically, 8.639 g of non-ground rose hips were transferred to a vacuum oven and dried for six hours at 85 °C. The weight of the dried samples was recorded, and moisture content was calculated using the following equation:(1)$$\;\%=\frac{loss\;in\;wt\times 100}{wt\;of\;sample}$$

#### 3.2.2. Extraction of Phenolics

In this step, 100 g of ground RH (previously treated with liquid nitrogen), both PEF-treated and CTRL samples, was added to 200 mL of either water or a water/ethanol mixture (50% *v*/*v*) and stirred in sealed bottles at 35 °C for 1 h. Subsequently, the samples were centrifuged and filtered. The resulting extracts were concentrated using a rotary evaporator to a final volume of 50 mL.

#### 3.2.3. Analysis of Polyphenols Without Fermentation

Then, 50 g of RH was dissolved in 60 mL of 0.1 N KOH solution. The solution was subjected to PEF treatment in a designated treatment cell. After PEF exposure, the solution was centrifuged at 8700 rpm for 30 min to separate the supernatant. A 20 mL aliquot of the supernatant was concentrated by vacuum evaporation to a final volume of 10 mL. Subsequently, 1 mL of the concentrated solution was filtered through a 0.20 μm microbial filter. The filtrate was then diluted with 4 mL of deionized water and prepared for LC-MS/MS analysis [30]. A control sample underwent the same procedure but omitted the PEF treatment.

#### 3.2.4. Analysis of Polyphenols After Fermentation

For polyphenol analysis, 12.5 g of RH were initially dissolved in 60 mL of 0.1 N KOH solution and treated with PEF in the treatment cell. Following PEF treatment, 10 mL aliquots from both the PEF-treated and control solutions were separately added to 100 mL of fermentation medium. Fermentation was conducted for 120 h under continuous stirring in a sealed 250 mL container. After fermentation, each solution was centrifuged at 8700 rpm. An 80 mL aliquot of the supernatant was collected and concentrated to a final volume of 10 mL. Finally, a 1 mL sample of each concentrated solution was filtered through a 0.20 μm microbial filter before analysis [30].

#### 3.2.5. Determination of Carotenoids

β-Carotene and lycopene were quantified following the procedure by Guimarães et al. [39]. The samples were placed in quartz cuvettes, and UV–Vis spectra were recorded. Absorbance was measured at 453, 505, 645, and 663 nm. The contents were calculated using the following equations:Lycopene (mg/10 mL) = −0.0458 × A_663_ + 0.204 × A_645_ + 0.372 × A_505_ − 0.0806 × A_453_(2)β-Carotene (mg/10 mL) = 0.216 × A_663_ − 1.220 × A_645_ − 0.304 × A_505_ + 0.452 × A_453_(3)

The results were expressed as mg of carotenoids per g of dry weight (DW).

#### 3.2.6. Fermentation Procedure

The process that began with the pretreatment of the RH was as follows: The fruits were washed, crushed, and pulped to release fermentable sugars and bioactive compounds. PEF treatment was applied prior to fermentation to improve polyphenol and sugar extraction, enhancing microbial activity. The fermentation medium consisted of RH juice or extract, with sugar content adjusted to 10–14 °Brix. Since *T. delbrueckii* has different nutrient requirements than *Saccharomyces*, a nitrogen source (e.g., diammonium phosphate) was added to support yeast metabolism. Each sugar—glucose, fructose, and maltose—was added at a concentration of 33 g/L in a nutrient-rich solution. The medium for the trials included 1 g/L yeast extract, 2 g/L (NH_4_)_2_SO_4_, 0.2 g/L MgSO_4_, 0.2 g/L ZnSO_4_, 1 g/L KH_2_PO_4_, and 1 g/L KHPO_4_, with pH adjusted to 4.5. Batches (1 L each) were transferred to sterilized flasks, autoclaved at 121 °C for 20 min, and inoculated with a mixed culture of *S. cerevisiae* and *T. delbrueckii* in a 1:10 ratio. Rose hips (100 g/L), both PEF-treated and control, were ground to a fine powder using a grinding bowl and added under sterile conditions. The fermentations were performed at 25 °C for 7 days. All trials were run in duplicate, and the results are presented as mean values. Fermentation proceeded under controlled conditions, beginning with an aerobic phase to support yeast growth, followed by an anaerobic phase to promote fermentation. The temperature was maintained at 20–25 °C and pH at 4–4.5, ensuring microbial stability and the preservation of sensory quality. The process typically lasted 5–10 days, during which *T. delbrueckii* metabolized residual sugars, producing esters, fruity volatiles, and organic acids that contributed to a floral, slightly tangy beverage profile. Upon completion, the beverage was filtered or centrifuged to remove yeast residues. Depending on the shelf-life requirements, the products were either stored at 4 °C for fresh consumption or pasteurized at 60–70 °C for longer stability. Optional carbonation could be introduced via secondary fermentation or CO_2_ injection.

#### 3.2.7. Volatile Compounds After Fermentation

The extraction method for volatile compounds produced during fermentation, as well as the instrumentation, column, and GC-MS conditions, were previously described by Drosou et al. [40]. Specifically, 100 g of ground RH (PEF-treated and control), previously cryogenically ground with liquid nitrogen, were added to 200 mL of ethanol solution (50% *v*/*v*) and stirred in sealed bottles at 35 °C for 1 h. The samples were then centrifuged and filtered. Extracts were concentrated using a rotary evaporator to a final volume of 50 mL. The samples were washed sequentially with three solvents (*n*-hexane, dichloromethane, and ethyl acetate) in a 1:2 ratio. The resulting organic phases were dried over anhydrous Na_2_SO_4_ and concentrated to 100 μL. A 1 μL aliquot was injected into the GC-MS for volatile compound analysis.

#### 3.2.8. Sugar Content

Analyses had been conducted using a UHPLC-MS (LCMS-2020, Shimadzu Corporation, Kyoto, Japan) coupled with a UV detector and a Mass Spectrometer detector. A HILIC Luna Omega Sugar column (Phenomenex, Torrence, CA, USA, 50 × 2.1 mm, 3 μm) was used. The system utilized electrospray ionization (ESI) in negative mode. The detection operated in SCAN and SIM modes. The optimized parameters included the following: DL temperature at 250 °C, nebulizer at 1.5 L/min, and dry gas at 10 L/min. The mobile phase consisted of H_2_O (Solvent A, UHPLC super gradient grade, Panreac AppliChem^®^) and acetonitrile (Solvent B, HPLC grade, Fisher Chemical). The gradient profile was as follows: initial 90% B for 6 min, a linear decrease to 70% B held for 1 min, then return to 90% B for 1 min, and equilibration at 90% B for 6 min, giving a total analysis time of 15 min. The flow rate was maintained at 0.25 mL/min, and the injection volume was 3 µL. Data acquisition spanned a mass-to-charge (*m*/*z*) of 100–1500, averaged and background-subtracted in ESI negative mode. Analysis was performed using LabSolution (5.113) software (Shimadzu Corporation, Kyoto, Japan). Standard sugar solutions (10 mg/L) were used to identify abundant ions: D(−)-Fructose (≥99%, Merck^®^) and D(+)-Glucose (≥97%, Roth^®^) deprotonated to form [M − H]^−^ ions, both detected at *m*/*z* 179.00. Their retention times were 2.1 and 2.71 min, respectively. SIM mode was preferred for targeted analysis. An additional sugar determination was performed using an enzymatic kit (Boehringer Mannheim/R-BIOPHARM).

#### 3.2.9. Liquid Chromatography Analysis

Liquid chromatography was performed using an Agilent 1100 Series system (vacuum degasser, UV VWD detector, binary pump) coupled to an LC/MSD Trap SL (Agilent Technologies Inc., Santa Clara, CA, USA). The column used was Purospher R STAR, RP-18 encapped (5 μm), Lichrocart 250-4. The mobile phase consisted of 0.1% (*v*/*v*) formic acid in water (solvent A) and MeOH (solvent B). The elution gradient was programmed as follows: initial 10% B, and a linear increase from 10% to 90% B until 40 min. The flow rate was maintained at 0.7 mL/min, and the injection volume was 20 µL. The UV detector operated at 280 nm. Mass spectrometric detection was conducted by operating in negative ion mode with an electrospray ionization (ESI) source (Agilent Technologies Inc., Santa Clara, CA, USA). The source-related parameters were optimized as follows: a desolvation temperature of 350 °C, nebulizer set at 40 psi, and dry gas at 9 L/min. Data were acquired over an *m*/*z* range of 100–1500, with a target mass of 594 *m*/*z* and compound stability of 100%.

Further analyses were performed using UHPLC-MS (LC-40DX3, Shimadzu Corporation, Kyoto, Japan) equipped with a UV detector and an RP-UHPLC-C18 column (Purospher STAR, RP-18, 2 μm). The ESI interface operated in negative ion mode. The mobile phase consisted of 0.1% (*v*/*v*) formic acid in water (solvent A) and MeOH (solvent B). The elution gradient was programmed as follows: initial 10% B, a linear increase from 10% to 90% B, hold at 90% B for 1 min, with a return and equilibration at 10% B from 21 to 25 min. The flow rate was maintained at 0.3 mL/min, and the injection volume was 10 µL. The UV detector operated at 280 nm. Mass spectrometric detection was conducted by operating in negative ion mode with an electrospray ionization (ESI) source (Shimadzu Corporation, Kyoto, Japan). The source-related parameters were optimized as follows: a DL temperature of 250 °C, nebulizer set at 1.5 L/min, and dry gas at 15 L/min. Data were acquired over an *m*/*z* range of 100–1500. Data were averaged and background-subtracted in negative ESI mode using LabSolution (ver. 5.113) software.

### 3.3. Statistical Analysis

All statistical analyses were conducted using XLSTAT Version 2024.3.0 (Addinsoft, Paris, France). Quantitative data are presented as mean ± standard deviation (SD) based on triplicate experiments unless otherwise specified. The normality of data distribution was assessed using the Shapiro–Wilk test. For the comparison of means between the control and PEF-treated groups, Student’s *t*-test was applied. Statistical significance was set at *p* < 0.05. When multiple compounds or conditions were compared simultaneously, one-way ANOVA followed by Tukey’s post hoc test was used. All *p*-values are reported explicitly in the results and corresponding tables to support the interpretation of significant differences.

## 4. Conclusions

This study confirmed the presence of key bioactive flavonoids in RH extracts, most notably tiliroside, robinin, catechin, and epicatechin. Tiliroside, detected at *m*/*z* 593.5 with a retention time of approximately 35 min, exhibited a distinct chromatographic peak that facilitated its confident identification. Its detection aligns with previous findings and reinforces its status as a major glycosylated flavonoid in rose hip matrices. Robinin, a kaempferol-based glycosyloxyflavone, was also identified, consistent with its well-documented antioxidant and anti-inflammatory properties.

Mass spectrometry analysis revealed catechin and epicatechin as monomers (*m*/*z* 289.5). While their exact isomeric forms require further confirmation, the data support their contribution to the flavonoid profile of *Rosa* species and their associated health benefits.

Comparative molecular abundance analysis revealed that fermentation significantly decreased levels of phenolic acids and flavonoids—especially citric/quinic acids and quercetin derivatives—likely due to microbial metabolism or compound degradation. PEF treatment, while exerting a more modest overall influence, enhanced the recovery of selected flavonoids such as phloridzin and macrulin 3C derivatives, especially in hydroalcoholic extracts post fermentation. These results suggest that PEF promotes the selective release of intracellular bioactives through cell membrane permeabilization.

Across both solvent systems, hydroalcoholic extraction yielded substantially higher molecular abundances than aqueous extraction, emphasizing the role of solvent polarity in optimizing phenolic compound recovery. Nevertheless, fermentation remained the predominant factor driving compound depletion, regardless of extraction strategy.

The manuscript discusses two mechanisms affecting phenolic compound extraction and stability during fermentation and processing, which merit emphasis as follows:

First, the apparent contradiction between the increased solubility of flavonols and polyphenols due to the alcohol produced during fermentation, and their overall reduction, can be explained by two concurrent processes: ethanol enhances solubility and extractability, but microbial enzymatic activity degrades or transforms these compounds, resulting in a net loss.

Second, the effectiveness of PEF treatment was contingent on solvent polarity. While PEF did not significantly improve extraction in water alone, it did so in hydroalcoholic mixtures. This likely reflects a synergistic effect, where ethanol’s intermediate polarity improves solubility and PEF enhances diffusion by disrupting cell membranes. Together, they facilitate the more efficient release of phenolics.

These insights highlight the complex interplay between solvent characteristics, fermentation dynamics, and electroporation-based extraction technologies in modulating phenolic compound stability and recovery.

In conclusion, this study underscores how solvent composition, fermentation parameters, and PEF treatment collectively shape the phytochemical profile of rose hip extracts. Given the antioxidant, anti-inflammatory, and potential anticancer properties of the detected flavonoids, the careful optimization of extraction protocols is crucial for preserving bioactive integrity in nutraceutical and pharmaceutical applications.

Future work should integrate tandem mass spectrometry and NMR spectroscopy for advanced structural elucidation, and include functional assays to investigate the biological activity and synergistic interactions of the recovered compounds.

## Figures and Tables

**Figure 1 molecules-30-03259-f001:**
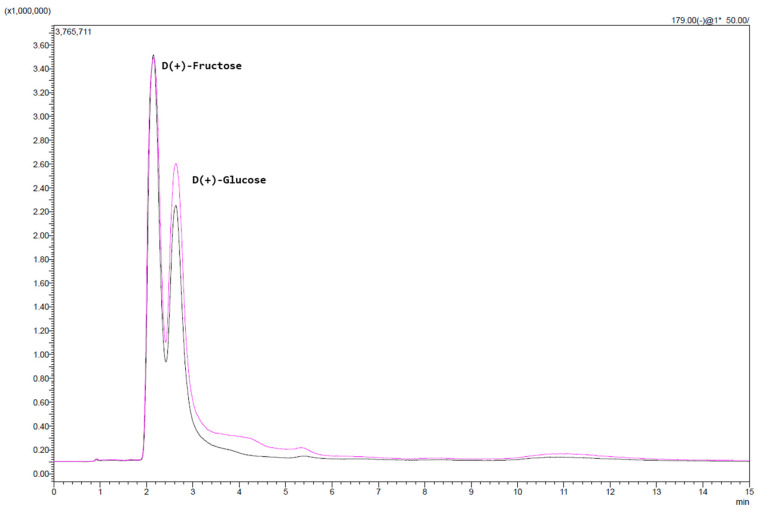
UHPLC-MS spectrum of sugars of PEF-treated (purple) sample and control (black).

**Figure 2 molecules-30-03259-f002:**
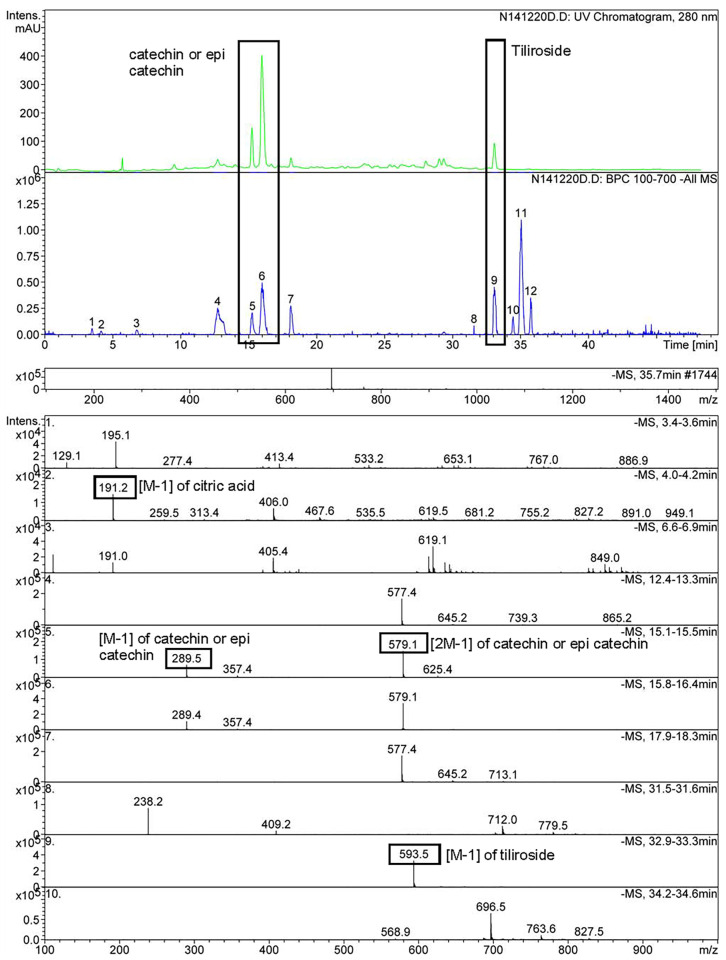
Chromatograms (UV, BPC) and mass spectra highlighting catechin, epicatechin, and tiliroside. Explanations of the numbered peaks are given in Table 5.

**Figure 3 molecules-30-03259-f003:**
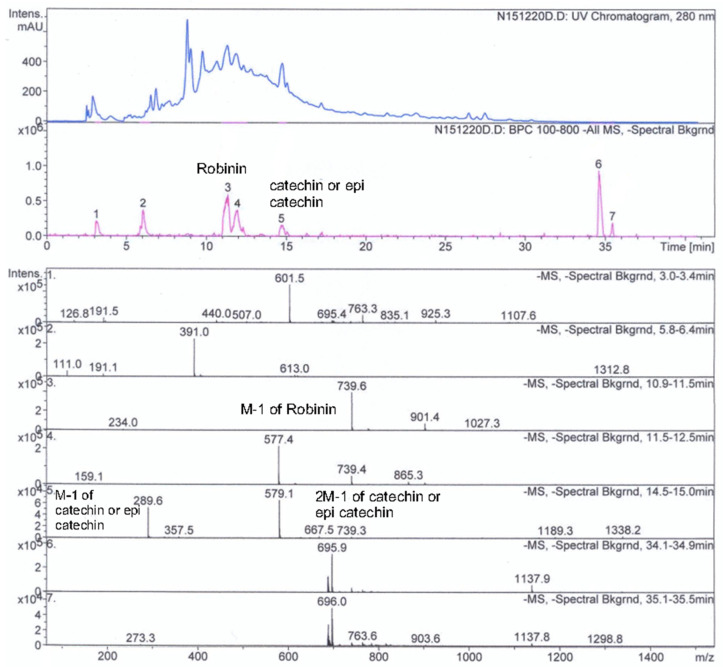
Chromatograms (UV, BPC) and mass spectra highlighting catechin, epicatechin, and robinin. Explanations of the numbered peaks are given in Table 5.

**Figure 4 molecules-30-03259-f004:**
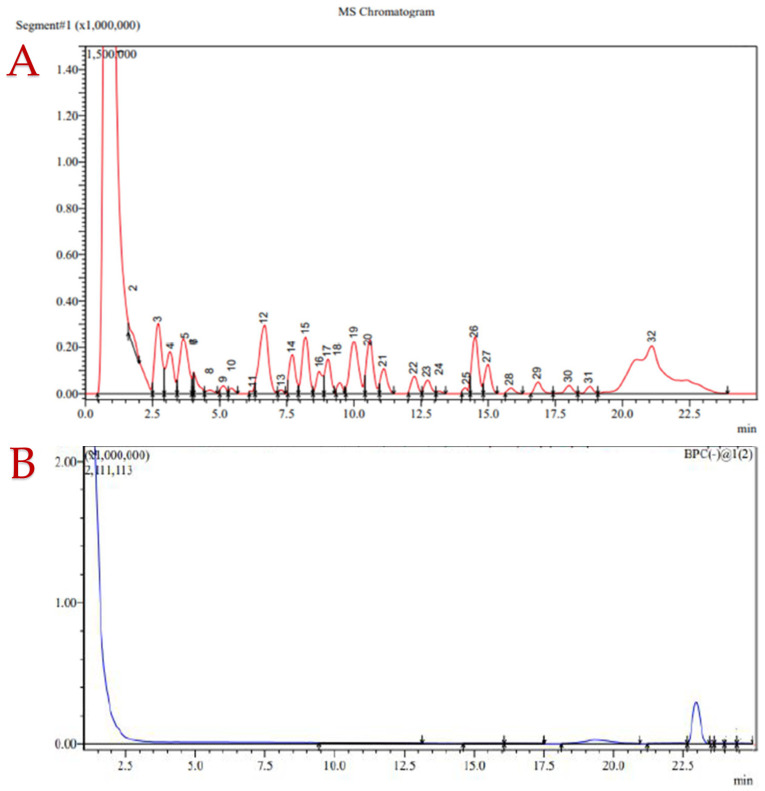
UHPLC/MS base peak chromatograms (BPC) showing plot (**A**) (red line) for the PEF sample and plot (**B**) (blue line) for the control. Explanations of the numbered peaks are given in Table 5.

**Figure 5 molecules-30-03259-f005:**
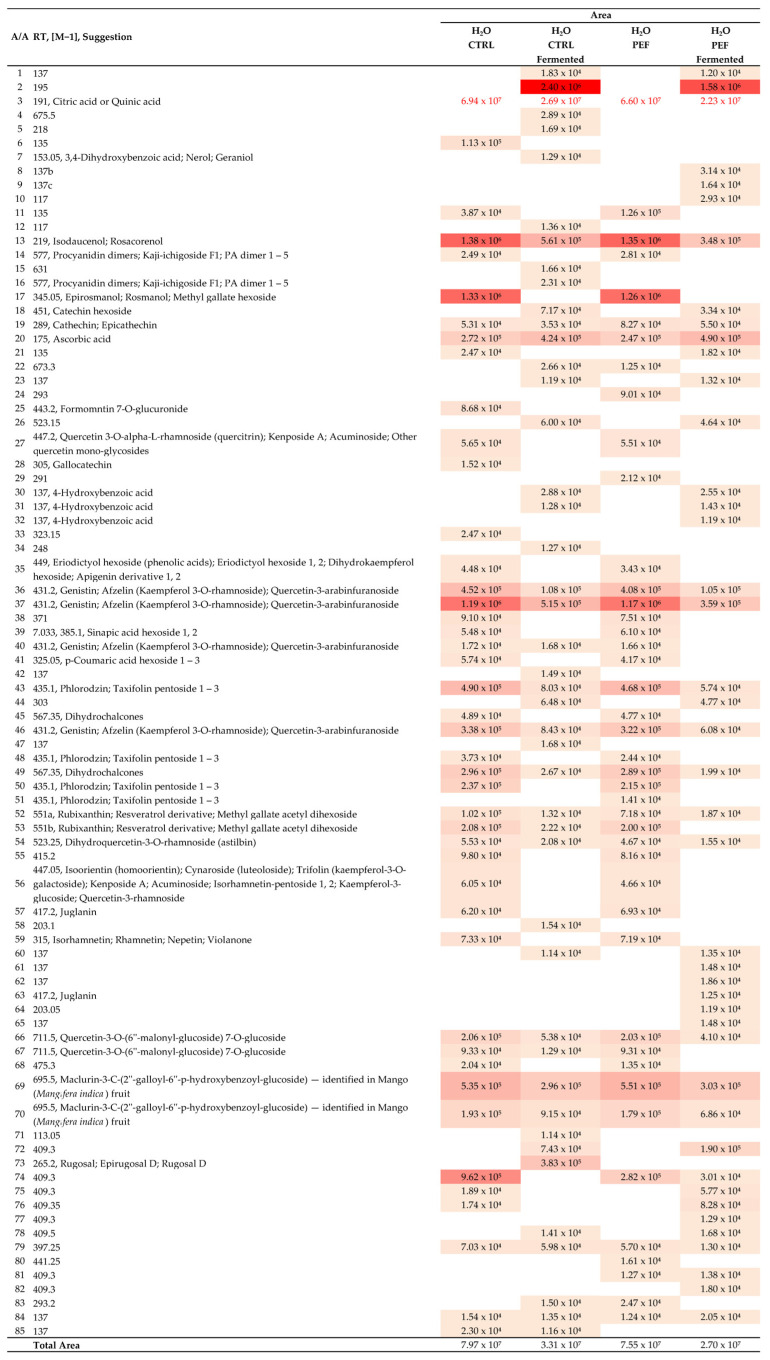
Heat map showing abundance variations in water extracts with control vs. PEF and fermented vs. non-fermented conditions. Cell colors range from light yellow (lowest intensity) to dark red (highest intensity).

**Figure 6 molecules-30-03259-f006:**
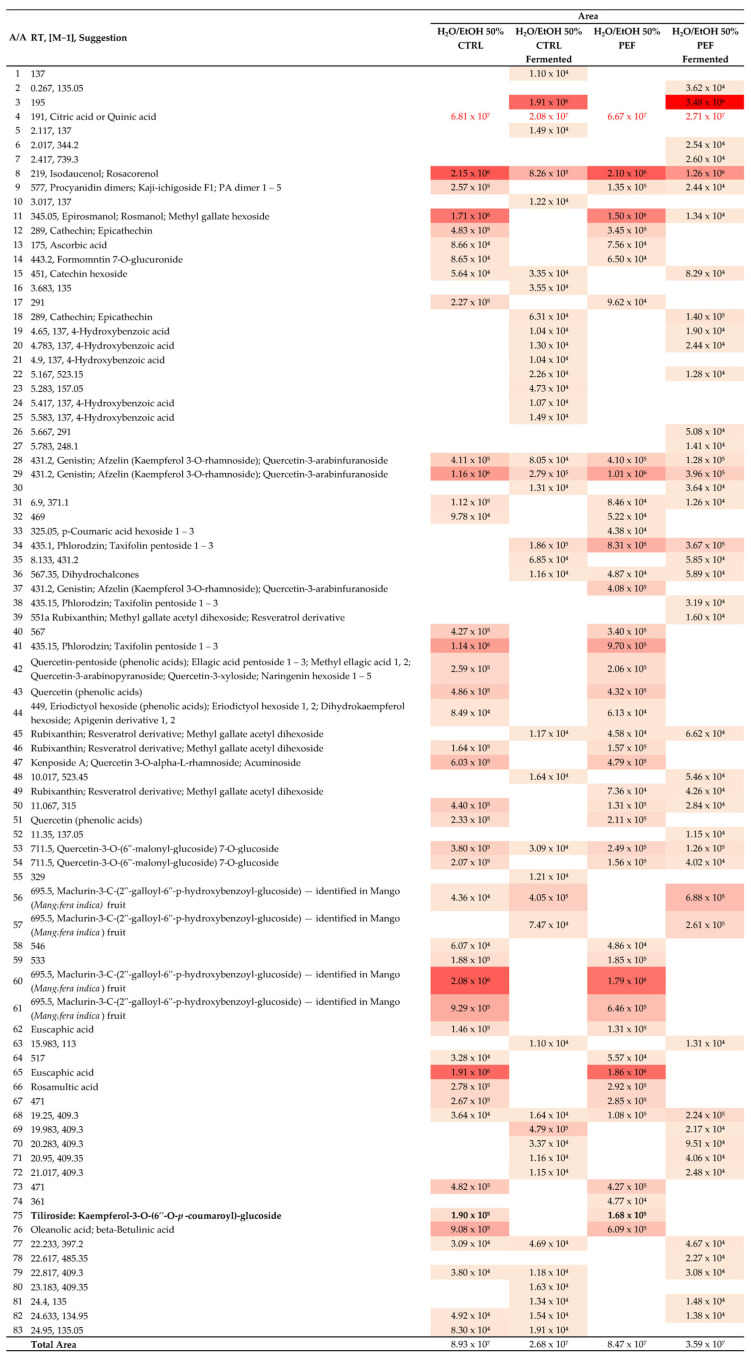
Heat map showing abundance variations in hydroalcoholic extracts with control vs. PEF and fermented vs. non-fermented conditions. Cell colors range from light yellow (lowest intensity) to dark red (highest intensity).

**Figure 7 molecules-30-03259-f007:**
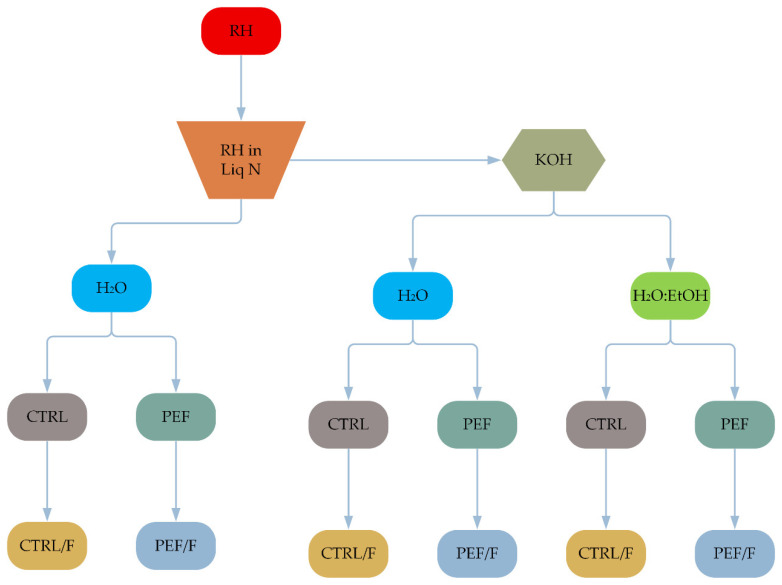
Schematic representation of the experimental design for rose hip (RH) treatment. The workflow includes two entry points: (1) direct dissolution in H_2_O solvent, (2) alkaline pretreatment with KOH (1 N), followed by dissolution in H_2_O or H_2_O:EtOH 50% *v*/*v* solvents. Each route branches into combinations of control (CTRL) or Pulsed Electric Field (PEF) treatment, with or without fermentation (F). The diagram visualizes the full factorial framework for analyzing the impact of pH alteration and fermentation on non-volatile compound stability.

**Table 1 molecules-30-03259-t001:** Main volatile compounds detected by GC-MS means at the end of fermentation in mg/L.

Compounds	CTRL	PEF
Isoamyl alcohol	10.30 ± 1.21^a^	11.19 ± 1.53 ^a^
Active amyl alcohol	4.24 ± 1.34 ^a^	3.86 ± 0.73 ^a^
Butanoic acid, 3-methyl-	0.11 ± 0.02 ^a^	0.11 ± 0.04 ^a^
Butanoic acid, 2-methyl-	0.11 ± 0.03 ^a^	0.11 ± 0.03 ^a^
1-Butanol, 3-methyl-, acetate	0.21 ± 0.00 ^a^	0.10 ± 0.01^b^
2H-Pyran-2,6(3H)-dione	0.20 ± 0.06 ^a^	0.14 ± 0.07 ^a^
Hexanoic acid	0.30 ± 0.20 ^a^	0.36 ± 0.09 ^a^
Hexanoic acid, ethyl ester	0.10 ± 0.01 ^a^	0.13 ± 0.11 ^a^
Phenylethyl Alcohol	5.96 ± 0.90 ^a^	8.85 ± 0.76 ^b^
Octanoic acid	1.05 ± 0.04 ^a^	1.40 ± 0.26 ^b^
Octanoic acid ethyl ester	0.13 ± 0.15 ^a^	0.26 ± 0.12 ^a^
Acetic acid, 2-phenylethyl ester	0.06 ± 0.03 ^a^	0.09 ± 0.03 ^a^
Nonanoic acid	0.03 ± 0.00 ^a^	0.03 ± 0.00 ^a^
α-Ionone	0.03 ± 0.00 ^a^	0.03 ± 0.00 ^a^
Eugenol	0.02 ± 0.00 ^a^	0.03 ± 0.01 ^a^
3-Decenoic acid, (E)-	0.21 ± 0.02 ^a^	0.09 ± 0.00 ^b^
n-Decanoic acid	0.16 ± 0.07 ^a^	0.22 ± 0.03 ^a^
Ethyl *trans*-4-decenoate	0.02 ± 0.00 ^a^	0.03 ± 0.01 ^a^
Decanoic acid, ethyl ester	0.01 ± 0.00 ^a^	0.03 ± 0.00 ^a^
Tyrosol	0.69 ± 0.10 ^a^	0.58 ± 0.04 ^a^
Vanillic acid	0.34 ± 0.02 ^a^	0.13 ± 0.07 ^b^
Tryptophol	0.33 ± 0.13 ^a^	0.76 ± 0.08 ^b^
1H-Indole-3-ethanol, acetate (ester)	ND	0.05 ± 0.00 ^a^
n-Hexadecanoic acid	0.53 ± 0.09 ^a^	0.90 ± 0.34 ^b^
Stearic acid	0.46 ± 0.09 ^a^	0.59 ± 0.16 ^a^

Different letters indicate statistically significant differences (Student’s *t*-test, *p* < 0.05). CTRL—control; PEF—Pulsed Electric Field; ND—Not Detected.

**Table 2 molecules-30-03259-t002:** Analysis of chemical compound changes: control vs. PEF treatment.

Category	Compound Type	Compound	Change Description	*p*-Value
Significant Increases	Aromatic Alcohols	Phenylethyl Alcohol	↑ 48.5% (from 5.96 to 8.85)	0.004
	Indole Derivatives	Tryptophol	↑ 130% (from 0.33 to 0.76)	0.002
	Long-chain Fatty Acids	*n*-Hexadecanoic Acid	↑ 69.8% (from 0.53 to 0.90)	0.047
Significant Decreases	Phenolic Acids	Vanillic Acid	↓ 61.8% (from 0.34 to 0.13)	0.011
	Unsaturated Acids	3-Decenoic Acid (*E*)	↓ 57.1% (from 0.21 to 0.09)	0.019
New Compounds	Esters	1H-Indole-3-ethanol, acetate	Detected only in PEF (0.05 ± 0.00)	-
Stable Compounds	Short-chain Acids	Butanoic Acid	No change (0.11 in both treatments)	0.615
	Simple Alcohols	Isoamyl Alcohol	Minimal change (from 10.30 to 11.19)	0.184
		Active Amyl Alcohol	Minimal change (from 4.24 to 3.86)	0.278

Individual *p*-values are provided to support statistical interpretation. ↑ indicates an increase relative to the control treatment, and ↓ indicates a decrease relative to the control treatment.

**Table 3 molecules-30-03259-t003:** Impact summary.

Metabolic Category	Observed Effects of PEF Treatment
Aromatic Compounds	Strong enhancement
Long-chain Fatty Acids	Moderate increase
Simple Alcohols	Minimal impact
Short-chain Acids	No significant change
Phenolic Compounds	Mixed responses (compound-dependent)
Esters	Slight overall increase

**Table 4 molecules-30-03259-t004:** Confidence-based grouping of key fermentation metabolites in PEF vs. CTRL treatments.

Confidence Level	Representative Compounds	Notes
High Reliability	Phenylethyl Alcohol, Tryptophol, Vanillic Acid	Low standard deviation; consistent across replicates
Moderate Reliability	*n*-Hexadecanoic Acid, Octanoic Acid	Acceptable variability; moderate standard deviation
Low Reliability	Various compounds with high standard deviation relative to mean	Greater variability; interpret with caution

All comparisons were evaluated using Student’s *t*-test. Significance was assessed at *p* < 0.05. Confidence categories were defined as follows: High (*p* < 0.05), Moderate (*p* ≈ 0.05), Low (*p* > 0.05).

**Table 5 molecules-30-03259-t005:** Suggestions for Figure 4 according to the literature and electronic databases [26,27,28,29,30].

Peak	Suggested Compound(s)	MW	Compound Class
1	Citric acid or Quinic acid	192	Organic acids
4	Procyanidin dimers (B1, B2, B3, B4, B5, B7)	740	Flavan-3-ol dimers (tannins)
5	Epirosmanol, Rosmanol	346	Phenolic diterpenoids
6	Catechin, Epicatechin	290	Flavan-3-ols
8	Hyacinthin, Pelargonin	595	Anthocyanins
10, 20	6-Hydroxyluteolin 7-O-rhamnoside, Homoorientin, Cynaroside, Trifolin, Astragalin, Quercitrin	448	Flavonoid glycosides
11, 19	Eriodictyol hexoside, Dihydrokaempferol hexoside, Apigenin derivative	450	Flavanone/flavonol glycosides
12, 13, 15	Quercetin-3-arabinofuranoside, Afzelin	431	Flavonol glycosides
14, 17, 18	Phloridzin (Phloretin 2′-O-glucoside)	435	Dihydrochalcone glycoside
16	Lutein, Epi-lutein	568	Carotenoids
24	Quercetin-acetylhexoside	506	Flavonol glycoside (acetylated)
30	Euscaphic acid, 3β,6α,19α-Trihydroxyurs-12-en-28-oic acid	488	Triterpenoid

**Table 6 molecules-30-03259-t006:** Key bioactive compounds detected in rose hips.

Compound Group	Key Constituents	Potential Bioactivities
Organic Acids	Citric Acid	Preservative, antioxidant, contributes tartness
Flavonoid Glycosides	Robinin, Kaempferol Glycosides, Quercetin Derivatives	Antioxidant, anti-inflammatory, potential anticancer
Procyanidins	Dimers B1–B7	Cardiovascular health, potent antioxidants
Diterpenes	Epirosmanol, Rosmanol	Antimicrobial, antioxidant
Flavan-3-ols	Catechin, Epicatechin	Cognitive and cardiovascular protection
Anthocyanins and Polyphenols	Cyanidin Glycosides, Vanillic Acid	Color compounds, antioxidants
Phytoestrogens and Isoflavones	Formononetin, Coumestrol, Genistin	Hormonal modulation, anti-inflammatory
Glycosylated Derivatives	Luteolin, Apigenin, Phloretin Variants	Anti-inflammatory, antioxidant, anticancer
Complex Polyphenols	Maclurin Derivatives, Phytenal	Signaling molecules, membrane components
Phenolic Acids	Protocatechuic Acid	Anti-inflammatory, neuroprotection

**Table 7 molecules-30-03259-t007:** Experimental groups for rose hip treatment combinations.

Group Code	Pretreatment	Solvent	Processing	Fermentation
RH/H_2_O/CTRL	None	H_2_O	CTRL	No
RH/H_2_O/PEF	None	H_2_O	PEF	No
RH/H_2_O/CTRL/F	None	H_2_O	CTRL	Yes
RH/H_2_O/PEF/F	None	H_2_O	PEF	Yes
RH/KOH/H_2_O/CTRL	KOH (1 N)	H_2_O	CTRL	No
RH/KOH/H_2_O/PEF	KOH (1 N)	H_2_O	PEF	No
RH/KOH/H_2_O/CTRL/F	KOH (1 N)	H_2_O	CTRL	Yes
RH/KOH/H_2_O/PEF/F	KOH (1 N)	H_2_O	PEF	Yes
RH/KOH/H_2_O:EtOH/CTRL	KOH (1 N)	H_2_O:EtOH	CTRL	No
RH/KOH/H_2_O:EtOH/PEF	KOH (1 N)	H_2_O:EtOH	PEF	No
RH/KOH/H_2_O:EtOH/CTRL/F	KOH (1 N)	H_2_O:EtOH	CTRL	Yes
RH/KOH/H_2_O:EtOH/PEF/F	KOH (1 N)	H_2_O:EtOH	PEF	Yes

RH—rose hip; CTRL—control; PEF—Pulsed Electric Field; F—fermentation.

## Data Availability

The original contributions presented in this study are included in the article. Further inquiries can be directed to the corresponding authors.

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
