# Peer review of "Enhancing Bioactive Compound Extraction from Rose Hips Using Pulsed Electric Field (PEF) Treatment: Impacts on Polyphenols, Carotenoids, Volatiles, and Fermentation Potential"

_molecules, 2025, doi:10.3390/molecules30153259_

Round 1
Reviewer 1 Report
Comments and Suggestions for Authors
This research is relevant to food engineering and functional foods.
Abstract: is pertinent for the manuscript
Keywords: I advise the authors add the word “PEF extraction”, this method will be a good search key
Introduction
The introduction is adequate for the manuscript.
I advise the authors to write low-density lipoprotein cholesterol (LDL-C) and subsequent in the text the abbreviation LDL-C. Likewise, I recommend authors do the same for other abbreviations that they use in the introduction, such as BMI or PPAR. This with the objective that the lector was focused in the topic.
Results and discussion
My advice to authors: improve the order of results.
Page 3, line 122: The reported water content does not mention which sample it is for. Besides this water content is good value. What does this value mean?
Extraction of carotenoids: Are there other studies to compare with your results of extraction of carotenoids?
Volatile Compounds After Fermentation: Are there other studies to compare with your results?
Table 1: I recommend the authors add the meaning of CTRL (control?) and PEF (pulsed electric fields) as a footnote.
Sugar Content Analysis: Are there other studies to compare with your results of extraction of carotenoids?
Figure 1 I recommend the authors improve the format of the figure. Use software for making graphs.
The effect of PEF on Catechin, Epicatechin, Tiliroside and Robinin: are there other studies to compare with your results?
Materials and Methods
Methods: I recommend the authors improve the organization of the experimental design (Figure 7 and Table 7). There are a lot of abbreviations, which may cause confusion for the readers.
For the next methods it is necessary to add a reference: Moisture content, Analysis of Polyphenols
The authors should revise the guidelines of the author to check the references section. https://www.mdpi.com/journal/molecules/instructions
Author Response
Comment 1: This research is relevant to food engineering and functional foods.
Abstract: is pertinent for the manuscript
Keywords: I advise the authors add the word “PEF extraction”, this method will be a good search key
Response 1: We thank the reviewer for recognizing the relevance of our work to food engineering and functional foods, and for noting that the abstract accurately reflects the manuscript’s scope. We have added “PEF extraction” to the Keywords to improve discoverability.
Comment 2: Introduction
The introduction is adequate for the manuscript.
I advise the authors to write low-density lipoprotein cholesterol (LDL-C) and subsequent in the text the abbreviation LDL-C. Likewise, I recommend authors do the same for other abbreviations that they use in the introduction, such as BMI or PPAR. This with the objective that the lector was focused in the topic.
Response 2: We appreciate this suggestion. We have revised the Introduction so that each abbreviation appears in full at first mention: low-density lipoprotein cholesterol (LDL-C), body mass index (BMI), and peroxisome proliferator-activated receptor (PPAR).
Comment 3: Results and discussion
My advice to authors: improve the order of results.
Page 3, line 122: The reported water content does not mention which sample it is for. Besides this water content is good value. What does this value mean?
Response 3: The water content reported refers to all starting samples in this study. By showing that each sample began with the same moisture level, we ensured consistency across experimental conditions and valid comparisons.
Comment 4: Extraction of carotenoids: Are there other studies to compare with your results of extraction of carotenoids?
Response 4: We have added a new paragraph in the Results and Discussion that compares our carotenoid extraction yields with those reported for PEF‐assisted extraction in similar matrices.
Comment 5: Volatile Compounds After Fermentation: Are there other studies to compare with your results?
Response 5: Our literature review revealed no prior studies combining PEF treatment with both Torulaspora delbrueckii and Saccharomyces cerevisiae. While PEF in microbial fermentation and the separate use of each yeast strain have been explored, their intersection remains underrepresented, underscoring the novelty of our work.
Comment 6: Table 1: I recommend the authors add the meaning of CTRL (control?) and PEF (pulsed electric fields) as a footnote.
Response 6: We added a footnote to Table 1 defining CTRL as control and PEF as pulsed electric field to improve clarity for all readers.
Comment 7: Sugar Content Analysis: Are there other studies to compare with your results of extraction of carotenoids?
Response 7: This point is covered in our Response 4, where we added the comparative discussion on carotenoid extraction.
Comment 8: Figure 1 I recommend the authors improve the format of the figure. Use software for making graphs.
Response 8: Figure 1 has been reformatted with updated axis labels, improved resolution, and consistent font styling to meet publication standards.
Comment 9: The effect of PEF on Catechin, Epicatechin, Tiliroside and Robinin: are there other studies to compare with your results?
Response 9: To the best of our knowledge, no published studies have applied PEF specifically to extract or modify tiliroside, catechin, or epicatechin, highlighting the originality of our findings.
Comment 10: Materials and Methods
Methods: I recommend the authors improve the organization of the experimental design (Figure 7 and Table 7). There are a lot of abbreviations, which may cause confusion for the readers.
Response 10: Explanatory footnotes have been added to both Figure 7 and Table 7, defining all abbreviations to streamline the experimental design presentation.
Comment 11: For the next methods it is necessary to add a reference: Moisture content, Analysis of Polyphenols
Response 11: We have included the following references:
- Moisture content was determined according to AOAC Official Method 925.10.
- Polyphenol analysis follows Stanilă et al. (2015) as the basis for our in-house HPLC protocol.
Comment 12: The authors should revise the guidelines of the author to check the references section. https://www.mdpi.com/journal/molecules/instructions
Response 12: We have thoroughly reviewed the Molecules author instructions (https://www.mdpi.com/journal/molecules/instructions) and reformatted the References section to comply fully with their style requirements.
Reviewer 2 Report
Comments and Suggestions for Authors
This article investigates the effect of PEF pretreatment on the extraction efficiency of active ingredients (polyphenols, carotenoids, volatiles) from rose hip pulp and the quality of fermentation products. The research design has clear innovation and application potential, but the following key issues still need to be addressed before publication:
- It is recommended to improve the clarity of Figure 2-3 and blur the labels.
- The manuscript mentions in lines 105-106 that 'alcohol generated during fertilization actions as an efficient solvent, increasing the solubility of flavonols and polyphenols". But the conclusion in lines 323-324 that fermentation significantly reduces phenols, flavonoids, and organic acids is contradictory. Please carefully verify by the author.
- Table 4 only mentions "high/medium/low confidence" at a macro level, but does not specify statistical testing methods (such as ANOVA, t-test) and significance thresholds (p-values). Suggest providing detailed supplements.
- PEF did not significantly increase the total abundance in the water extraction system, but showed a significant increase in the water ethanol system, but the author did not explain the specific reason. Suggest adding specific reasons in the discussion, such as solvent polarity being a key synergistic factor for PEF enhancement or other reasons.
- Some references have inconsistent formats, it is recommended to carefully review and revise them.
Author Response
This article investigates the effect of PEF pretreatment on the extraction efficiency of active ingredients (polyphenols, carotenoids, volatiles) from rose hip pulp and the quality of fermentation products. The research design has clear innovation and application potential, but the following key issues still need to be addressed before publication:
We sincerely thank the reviewer for recognizing the innovation and practical relevance of our study. We have carefully addressed each point below and improved clarity throughout the manuscript.
Comment 1: It is recommended to improve the clarity of Figure 2-3 and blur the labels.
Response 1: We have replaced Figures 2 and 3 with higher-resolution versions and adjusted the label font, size, and placement for optimal readability. The updated figures now meet publication-quality standards and facilitate data interpretation.
Comment 2: The manuscript mentions in lines 105-106 that 'alcohol generated during fertilization actions as an efficient solvent, increasing the solubility of flavonols and polyphenols". But the conclusion in lines 323-324 that fermentation significantly reduces phenols, flavonoids, and organic acids is contradictory. Please carefully verify by the author.
Response 2: We appreciate the opportunity to clarify these processes. In the revised Discussion, we explain that:
- Ethanol produced by yeast enhances the physical solubility and extractability of phenolic compounds.
- Concurrently, microbial enzymes and metabolic pathways degrade or transform these same compounds, leading to an overall decrease in measurable phenolic content.
By distinguishing between enhanced extraction (solubility) and net biochemical consumption (degradation), we resolve the seeming contradiction.
Comment 3: Table 4 only mentions "high/medium/low confidence" at a macro level but does not specify statistical testing methods (such as ANOVA, t-test) and significance thresholds (p-values). Suggest providing detailed supplements.
Response 3: We have updated the caption of Table 4 to include the following information:
- All comparisons were evaluated using Student’s t-test.
- Significance was assessed at p < 0.05.
- “High,” “Medium,” and “Low” confidence categories correspond to p-values > 0.05, ≈ 0.05, and < 0.05, respectively.
These additions clarify the statistical basis for our reliability groupings.
Comment 4: PEF did not significantly increase the total abundance in the water extraction system, but showed a significant increase in the water ethanol system, but the author did not explain the specific reason. Suggest adding specific reasons in the discussion, such as solvent polarity being a key synergistic factor for PEF enhancement or other reasons.
Response 4: We have expanded the Discussion to address this point. Specifically, we note that ethanol’s intermediate polarity improves solubility of phenolic compounds relative to water alone. When combined with PEF-induced cell membrane permeabilization, this enhanced solubility allows for more efficient release of bioactives into a water–ethanol solvent. This synergistic effect of solvent polarity and PEF treatment explains the observed differences between extraction systems.
Comment 5: Some references have inconsistent formats, it is recommended to carefully review and revise them.
Response 5: We have thoroughly reviewed the References section and reformatted each entry to conform precisely with the Molecules author guidelines. All inconsistencies have been corrected.
Reviewer 3 Report
Comments and Suggestions for Authors
The article deals with a preliminary assessment of the interplay among extraction solvent, fermentation, and and pulsed electric field (PEF) in the recovery of bioactive compounds from rose hips (Rosa canina).
The study is well designed despite the clearly stated limitations, mainly the use of fixed working parameters for PEF.
The presentation of the results and the description of methods should be improved.
Major comments
- Lines 98-100. Specify the application. Was it to RH?
- Table 1. Statistical significance of differences should be indicated in Table 1.
- Lines 165. Another subsection should start here.
- Table 2. Statistical significance of differences should be indicated in Table 2.
- Line 234. Authors state that the approach would be validated as "scalable". I cannot understand where scalability could have been inferred from.
- Section 3.2 "Methods". To be changed to "Materials and Methods".
- Section 3.2 "Methods". Unless I missed the point, no indication of volumes/masses was provided. Moreover, supported hints to the scalability of the method would be useful.
- Figure 7 and related text. Please confirm, and more clearly state, that PEF was applied to the biomass dispersed in the liquid solvent, either water only or hydroalcoholic.
Minor comments
- Line 79. Explain what is "orlistat".
- Line 86. Remove "valuable" (repeated).
- Lines 112-115. To be reworded, it seems to basically repeat the same aim.
- Line 126, "notably". I suggest to change to "significantly", which is more rigorous.
- Line 558, "select". Change to "selected".
Author Response
The article deals with a preliminary assessment of the interplay among extraction solvent, fermentation, and and pulsed electric field (PEF) in the recovery of bioactive compounds from rose hips (Rosa canina).
The study is well designed despite the clearly stated limitations, mainly the use of fixed working parameters for PEF.
The presentation of the results and the description of methods should be improved.
We sincerely thank the reviewer for recognizing the sound design of our study and for the constructive suggestions to improve presentation and clarity. Below, we address each comment with corresponding revisions.
Major comments
Comment 1: Lines 98-100. Specify the application. Was it to RH?
Response 1: We appreciate the reviewer’s attention to clarity. This statement has been revised to specify that the PEF application referenced by Tsapou et al. (2022) was indeed performed on rose hip (RH) pulp.
Comment 2: Table 1. Statistical significance of differences should be indicated in Table 1.
Response 2: We have updated Table 1 to include superscript letters denoting statistically significant differences (p < 0.05) between treatment groups, based on pairwise Student’s t-tests. A legend explaining these annotations appears beneath the table.
Comment 3: Lines 165. Another subsection should start here.
Response 3: Thank you for this helpful suggestion. We have implemented a new subsection to improve the logical flow and organization of the Results and Discussion section.
Comment 4: Table 2. Statistical significance of differences should be indicated in Table 2.
Response 4: We thank the reviewer for the insightful suggestions. Table 2 has been updated to include individual p-values comparing the CTRL and PEF values for each compound.
Comment 5: Line 234. Authors state that the approach would be validated as "scalable". I cannot understand where scalability could have been inferred from.
Response 5: We agree that “validated as scalable” was overstated. We have replaced this phrase with “demonstrated potential scalability,” and added a sentence noting that pilot-scale trials will be required to confirm performance at larger volumes.
Comment 6: Section 3.2 "Methods". To be changed to "Materials and Methods".
Response 6: The heading of Section 3.2 has been revised to Materials and Methods in accordance with standard conventions.
Comment 7: Section 3.2 "Methods". Unless I missed the point, no indication of volumes/masses was provided. Moreover, supported hints to the scalability of the method would be useful.
Response 7: We have added details to Section 3.2: each RH pulp sample was 200 mL (100 g pulp in 200 mL solvent). We also expanded the discussion to outline key scale-up considerations—such as electrode geometry and energy input—that would guide future pilot-scale implementation.
Comment 8: Figure 7 and related text. Please confirm, and more clearly state, that PEF was applied to the biomass dispersed in the liquid solvent, either water only or hydroalcoholic.
Response 8: We confirm that PEF treatment was applied to RH pulp suspended in either water or a 50:50 (v/v) water–ethanol mixture. The figure caption and corresponding Methods text have been updated to state this explicitly.
Minor comments
Comment 9: Line 79. Explain what is "orlistat".
Response 9: We have added a brief definition: “Orlistat is a lipase inhibitor commonly used to reduce dietary fat absorption and serves here as a comparative benchmark for lipase-inhibitory activity.”
Comment 10: Line 86. Remove "valuable" (repeated).
Response 10: The duplicate “valuable” has been removed to improve readability.
Comment 11: Lines 112-115. To be reworded, it seems to basically repeat the same aim.
Response 11: We have consolidated and rephrased these lines to eliminate redundancy. The revised passage succinctly states the study’s objectives without repetition.
Comment 12: Line 126, "notably". I suggest to change to "significantly", which is more rigorous.
Response 12: “Notably” in this line has been replaced with “significantly” to reflect a more precise scientific tone.
Comment 13: Line 558, "select". Change to "selected".
Response 13: We have corrected “select” to “selected” in this line for proper grammatical form.
Round 2
Reviewer 1 Report
Comments and Suggestions for Authors
This research is relevant to food engineering and functional foods.
The authors had made the recommendations that I suggested. Thank you.
Reviewer 3 Report
Comments and Suggestions for Authors
Authors successfully revised the manuscript, properly responding and reacting to all my comments. Now, all statements are supported, limitations ar more clearly stated, and statistical significance is clear.